# Combination Therapy to Treat Fungal Biofilm-Based Infections

**DOI:** 10.3390/ijms21228873

**Published:** 2020-11-23

**Authors:** Jana Tits, Bruno P. A. Cammue, Karin Thevissen

**Affiliations:** Centre of Microbial and Plant Genetics, Katholieke Universiteit Leuven, 3001 Leuven, Belgium; jana.tits@kuleuven.be (J.T.); bruno.cammue@kuleuven.be (B.P.A.C.)

**Keywords:** biofilms, combination treatment, potentiator, synergy, mode of action, *Candida*, *Aspergillus*, *Cryptococcus*, virulence factors, tolerance mechanisms

## Abstract

An increasing number of people is affected by fungal biofilm-based infections, which are resistant to the majority of currently-used antifungal drugs. Such infections are often caused by species from the genera *Candida, Aspergillus* or *Cryptococcus.* Only a few antifungal drugs, including echinocandins and liposomal formulations of amphotericin B, are available to treat such biofilm-based fungal infections. This review discusses combination therapy as a novel antibiofilm strategy. More specifically, in vitro methods to discover new antibiofilm combinations will be discussed. Furthermore, an overview of the main modes of action of promising antibiofilm combination treatments will be provided as this knowledge may facilitate the optimization of existing antibiofilm combinations or the development of new ones with a similar mode of action.

## 1. Introduction

Various bacterial and fungal species form biofilms upon adherence to biotic or abiotic surfaces [1,2]. Biofilms are organized microbial populations embedded in a self-produced extracellular polymer matrix and can be prevalent in natural, industrial, and hospital settings [1,3,4,5]. Microbial cells within a biofilm are physiologically distinct from planktonic cells of the same organism and are generally highly tolerant to current antimicrobial drug classes [4,6,7]. In addition, they are protected from the host’s immune system [8,9]. Consequently, microbial biofilms are difficult to eradicate and are regarded as a prevalent cause of treatment failure and infection recurrence [10,11,12]. The U.S. National Institutes of Health estimated that 80% of all microbial infections in the human body are biofilm-based [13,14]. Bacterial and fungal biofilms can cause infections of various body sites, such as reproductive organs, the respiratory—and urinary tract and the oral cavity, as well as on implanted medical devices [2,15].

In this review, we will elaborate on the potential of combination therapy in the treatment of fungal biofilm-based infections. Over the past decades many studies have been performed to investigate the potential of combining compounds to treat fungal infections. In this respect, antifungal drug combinations voriconazole-echinocandins and flucytosine-fluconazole were identified as promising treatments against invasive aspergillosis and cryptococcosis, respectively (reviewed by [16]). Although the fungal biofilm state is the main cause of treatment failure and recurrence, mere antifungal activity of combinations is often assessed in a planktonic rather than in a biofilm setup. Therefore, this review will focus on the more recent literature regarding the combination of an antifungal with a non-antifungal, a so-called potentiator, that is able to enhance the activity of the antifungal drug against fungal biofilms, and will leave combinations of antifungal drugs out of consideration (reviewed by [16]). Such combination therapy based on potentiation is expected to result in a widened drug activity spectrum, lower doses of toxic drugs, quicker antifungal action and a lower risk for the occurrence of fungal drug resistance [17]. Most studies regarding antibiofilm combinations have been performed on *Candida albicans*. Hence, this review mainly focuses on antibiofilm combinations against *C. albicans* although we have included relevant info on other pathogens whenever available.

## 2. Fungal Biofilm-Based Infections: Problem and Current Therapeutic Options

It is estimated that fungal infections, especially those caused by yeasts of the genera *Candida* or *Cryptococcus* or by airborne filamentous fungi of the genus *Aspergillus*, result in more than one million deaths each year [18]. An increasing number of people is susceptible to such infections due to the rising number of immunocompromised individuals and the augmented use of implanted medical devices, which are potential substrates for biofilms [19,20,21,22,23]. *Candida* spp., of which *C. albicans* is most common, are involved in 60% of all opportunistic mycoses and systemic *Candida* infections result in 150,000 deaths annually [24]. *Aspergillus fumigatus* is the most prevalent *Aspergillus* spp. and can cause opportunistic cutaneous infections and allergic or chronic airway infections [25,26]. Moreover, severe invasive infections with high mortality rates can occur [25]. Finally, *Cryptococcus* spp. of which *Cryptococcus neoformans* and *Cryptococcus gattii* are the most common species involved in human diseases, mainly target the central nervous system but can also cause pulmonary or cutaneous infections [27,28].

Hence, medically important fungal species can infect various parts of the human body. For instance, the female reproductive organ is sensitive to infections by both bacterial and fungal species. Bacterial vaginosis is the most common vaginal infection and is generally accepted to involve multi-species vaginal biofilms that contribute to its pathogenesis [29,30,31,32]. Although the importance of biofilms in vulvovaginal candidiasis has been the subject of much debate, an increasing amount of evidence exists pointing to an important role of biofilms in pathogenesis and treatment failure of this infection [10,33,34,35]. Even though in vivo biofilm formation is most extensively studied for *C. albicans,* biofilm formation by *A. fumigatus* has been demonstrated in vivo and evidence suggests that common *Aspergillus* infections, such as aspergilloma and invasive pulmonary aspergillosis, are biofilm-based [36]. Furthermore, Wu and coworkers found that the formation of *C. albicans* biofilms in vulvovaginal candidiasis stimulates the occurrence of persister cells, which are residual biofilm cells that are tolerant to high antifungal doses and possess the capacity to grow upon removal of the antifungal treatment, resulting in new biofilms [33,37]. Therefore, these researchers suggested the formation of persister cells as an important antimicrobial tolerance mechanism of vaginal *C. albicans* biofilms, which is also the case for various bacterial biofilm-based infections [33,38,39]. Oral candidosis is also biofilm-based, typically involving mixed candida-bacterial biofilms [22]. Various studies point to the importance of such polymicrobial biofilms as they can elicit a more severe inflammatory response and/or are characterized by increased antifungal resistance as compared to monospecies biofilms in vivo [40,41]. Moreover, biofilms can form on indwelling medical devices, like catheters, and develop into severe recurrent systemic infections with high mortality rates when fungal cells enter the bloodstream [42,43,44].

Four major classes of antimycotics are currently used to treat patients suffering from fungal infections [45]. Polyenes, like amphotericin B (AmB), are the oldest group of antifungal drugs. AmB is characterized by a fungicidal action [46,47]. AmB binds to membrane sterols to form transmembrane pores, thereby altering membrane permeability and causing leakage of essential components out of the cell [48,49]. Additionally, it induces oxidative damage [47]. However, several different models have been proposed for its mode of action. In contrast to the generally accepted pore model, Anderson and coworkers introduced the sterol sponge model in which AmB forms extramembranous and fungicidal clusters that extract ergosterol from lipid bilayers [50]. Moreover, recent fluorescence lifetime imaging microscopy data indicate that AmB is characterized by multiple mechanisms of action as extra-membranous sponge-like structures, drug binding into fungal membranes and intracellular AmB-containing vesicular structures that probably target various organelles have all been observed in AmB-treated *C. albicans* cells [51]. AmB is the preferred treatment for systemic fungal infections and one of its main side effects, being nephrotoxicity, could be reduced but not eliminated upon its liposomal formulation [52,53,54]. A second antifungal class are azoles, which act by inhibiting the enzyme lanosterol 14-alpha-demethylase [55,56]. This enzyme is important for the biosynthesis of ergosterol, a major compound of the fungal membrane [57]. Azoles are the preferred topical treatment for vulvovaginal candidiasis and cutaneous fungal infections as they are relatively inexpensive and have a limited toxicity [58]. However, azoles have only moderate antibiofilm activity and resistance is occurring [58,59,60]. Third, echinocandins (e.g., caspofungin) inhibit β-1,3-d-glucan synthase, a crucial enzyme in the synthesis of β-1,3-glucan [61,62]. The latter is a major constituent of the fungal cell wall and occurs in the extracellular polymeric matrix of biofilms of various fungal species [63,64,65]. Echinocandins are fungicidal against yeasts, but fungistatic against moulds [66]. Furthermore, they are embryotoxic and have a limited activity against, for example, *C. neoformans* [67,68]. Finally, allylamines (e.g., terbinafine) interfere with ergosterol biosynthesis by inhibiting the enzyme squalene epoxidase [69,70]. Terbinafine, for example, is often used to treat superficial fungal infections of the nails, skin and hair [71]. In vivo it is active against dermatophytes, but, however, barely against *Candida* or mold species [72,73,74]. Out of this antifungal armamentarium only echinocandins and liposomal formulations of AmB can be used to treat biofilm-based infections [75,76,77]. Hence, there is a strong need for development of novel and more effective antibiofilm treatment strategies.

There are several approaches to develop new antibiofilm treatments, ranging from the development of new chemical entities that act on novel selective biofilm targets, to modifying existing antifungal drugs or repurposing existing drugs, primarily used to treat conditions other than fungal infections. A chemical modification on the echinocandin backbone for example resulted in an increased stability of echinocandin CD101, allowing the use of this antibiofilm drug as a topical agent for skin and vaginal infections [78,79]. Furthermore, several studies suggested antimicrobial photodynamic therapy as a promising method to treat biofilm-based infections caused by for example *C. albicans* [80,81,82]. Important virulence factors, such as biofilm formation and adhesion ability, can be affected using this method [82,83]. Briefly, a photosensitizer is applied and excited by harmless light at a specific wavelength, resulting in the generation of reactive oxygen species in the presence of oxygen molecules and consequently in cell damage, cell death or for example destruction of the biofilm extracellular matrix [80]. Moreover, application of nanoparticles as a new drug delivery system to avoid, for example, off-target side effects is currently investigated as well as the use of antifungal lock therapy to address the problem of catheter-related bloodstream infections [84,85]. Here, high concentrations of antimicrobials are instilled into the catheter lumen on which biofilms were formed [84]. Another approach is repurposing of existing drugs, primarily used to treat conditions other than fungal infections. Apparently, auranofin, commonly used to treat rheumatoid arthritis, is active against *Candida* biofilms in vitro and proven capable of reducing fungal burden in vivo in the model organism *Caenorhabditis elegans* infected with *C. neoformans* [86]. In general, *C. elegans* is an interesting in vivo model organism to study fungal infections caused by several medically important fungal pathogens (reviewed by [87]). Several virulence factors, such as hyphal formation or the polysaccharide capsule have been identified in *C. elegans* infected with *C. albicans* or *C. neoformans*, respectively [87]. Furthermore, various studies describe *C. elegans* as a model organism that enables high throughput screening for preclinical drug discovery, especially against infection by *C. albicans* [88,89]. The use of *C. elegans* as an in vivo model for *A. fumigatus* infections is still under investigation, but promising results have recently been reported regarding pathogenicity investigation, screening for new antifungal drugs and assessment of drug efficacy [90].

## 3. The Search for Synergistic Antibiofilm Combinations

### 3.1. Drug Interactions: Synergy

The search for potent antibiofilm combination therapies starts by determining the nature of the compounds’ interactions in vitro. To this end, a checkerboard assay [91] is typically used. Here, a combination of the antifungal drug and the potentiator, two-fold diluted across rows and columns of a microplate, respectively, is added to the fungal biofilms, followed by quantification of the biofilms by cell viability staining (e.g., CellTiter-Blue or XTT staining) [91,92,93,94,95]. The resulting data can be analyzed by various approaches that have been described, reviewed and compared in literature [96,97,98,99]. Two frequently used approaches are the calculation of the Fractional Inhibitory Concentration Index (FICI) [91], based on the Loewe Additivity Model [100], and the ΔE model [99,101], based on the Bliss Independence Theory [102]. In these two models, the assumption of no interaction is central as interactions are termed synergistic or antagonistic based on the deviation from the state of no interaction [91,99]. In a biofilm eradication or inhibition setup, the formula for the FICI is made up of different biofilm eradication concentration 2 (BEC-2) values or biofilm inhibition concentration 2 (BIC-2) values, defined as the minimal compound concentration resulting in 50% mature biofilm eradication or in a 2-fold inhibition of biofilm formation, respectively [103]. Assuming a biofilm eradication setup, FICI = [C(BEC-2A)/BEC-2A] + [C(BEC-2B)/BEC-2B]. Here, C(BEC-2A) and C(BEC-2B) represent the BEC-2 values of antifungals in combination and BEC-2A and BEC-2B the BEC-2 values of antifungals A and B on their own. The interaction is termed synergistic, indifferent or antagonistic if FICI ≤ 0.5, 0.5 < FICI < 4 or FICI ≥ 4, respectively [91,103]. FICI calculations are simple and feasible, but different definitions are described in literature, resulting in different outcomes and interpretation of results [91,99,104]. The ΔE model is a newer method in which ΔE stands for the difference between the predicted and measured growth percentages with drugs at several concentrations. More specific, ΔE = E_predicted_ − E_measured_ and E_predicted_ = E_A_ × E_B_ in which E_A_ and E_B_ are the measured growth percentages when drugs A and B act independently. The interaction of the drugs at the specific tested concentrations is considered synergistic or antagonistic if ΔE is positive or negative, respectively, with the 95% confidence interval excluding 0. Otherwise, Bliss independence is concluded. Typically, the sum percentages of all significant synergistic and antagonistic interactions are calculated and interactions are termed weak, moderate or strong if the percentage statistically significant interactions amounts <100%, 100–200% or >200%, respectively [98,99]. In general, results obtained by FICI calculations and ΔE are similar, although the ΔE method tends to be more consistent [99,105]. In addition to the checkerboard assay, time-kill studies can provide insights into a combination’s rate and extent of antibiofilm activity over time as compared to single compound or control treatments. Here, biofilms of different treatment groups are quantified at various time points during treatment by means of cell viability staining (e.g., XTT staining) or CFU determination [105,106,107,108]. Various other methods to determine in vitro compound interactions exist, like a disk diffusion-based assay [109,110,111] or Etests [112,113], but those techniques are only suited to assess combinations’ effects on planktonic cultures and not specific on biofilms.

Different methods can lead to different results and all have advantages and drawbacks. For example, a major disadvantage of simple and feasible FICI calculation is its sensitivity to intra-experimental errors [98,99], while the ΔE method is often regarded as more consistent [99,105]. However, the latter method is criticized as it does not exclude that a drug interacts with itself, resulting in synergy [101]. Therefore, both methods were used in several studies in addition to time-kill experiments, providing insights into the combination’s antibiofilm activity over time [95,99,105]. Combining different models/methods allows for the most accurate prediction of the clinical outcome of combination therapy.

Examples of synergistic antibiofilm combinations in vitro are AmB combined with the anti-inflammatory drug aspirin and triazoles in combination with histone deacetylase inhibitor vorinostat against biofilms of *Candida parapsilosis* and *C. albicans*, and against several *Aspergillus* spp. biofilms, respectively [105,114]. These combinations were tested against biofilms based on the performance against planktonic fungal cultures, the (weak) antibiofilm activity or based on previously observed synergy between similar compounds (e.g., histone deacetylase inhibitors other than vorinostat) and the antifungal drug. Furthermore, a recent study identified a synergistic combination consisting of the antibiotic minocycline and fluconazole against early-stage biofilms of susceptible and resistant *C. neoformans* strains, but not against mature biofilms [115]. Examples of antibiofilm combinations identified by means of more systematic screening will be discussed in the next section.

### 3.2. Screening for Novel Antibiofilm Combinations

In general, novel potentiators of currently used antifungal drugs are discovered by ad random screening of various types of small molecule or peptide libraries in combination with sub-optimal doses of the antifungal drug [92,116,117,118]. On the other hand, novel potentiators can also be identified using bio-informatic tools that integrate knowledge regarding previously discovered combinations or by focusing on inhibiting biofilm-specific tolerance mechanisms [119,120].

Various potentiators have been discovered by screening drug repurposing libraries, composed of off-patent drugs and bioactive agents with a safe toxicity profile and known dosing regimens. Repurposing is advantageous as possible drug reformulations are associated with lower costs as compared to designing a completely new drug [121,122]. Briefly, repurposing libraries are screened by treating fungal biofilms with sub-optimal antifungal drug doses combined with a library compound, followed by cell viability staining and validation experiments to determine the most optimal combination treatment (e.g., based on synergy or fungicidal activity as described in Section 3.1). Such a setup was, for example, used by De Cremer and colleagues who screened 1600 compounds and identified antimalarial artemisinins as novel miconazole potentiators against *C. albicans* biofilms [92]. Using a similar setup, we recently identified the quaternary ammonium compound domiphen bromide as another miconazole potentiator by screening 1311 additional repurposing compounds [118]. We showed that domiphen bromide enhanced miconazole’s antibiofilm activity against susceptible *C. albicans* as well as fluconazole-resistant *C. albicans* isolates, intrinsically azole-resistant *Candida glabrata* and against the emerging pathogen *Candida auris* [118]. In addition, Lafleur and coworkers screened a total of more than 60,000 compounds from several libraries (Asinex 1, ChemBridge 3, ChemDiv 3, ChemDiv 4, Enamine 2, Maybridge 5) and identified 2-adamantanamine as an azole potentiator against *C. albicans* biofilms [117]. 2-adamantanamine is a derivative of the anti-influenza A drug amantadine, which is also used in treatment of Parkinson’s disease symptoms [123]. Also potentiators of other antifungal classes have been identified. Toremifene citrate, normally used in the treatment of breast cancer, as well as the progestin drospirenone and the anti-anginal drug perhexiline maleate appeared capable of potentiating both AmB and caspofungin against *C. albicans* and intrinsically azole-resistant *C. glabrata* biofilms in a similar screen of 1600 compounds [116].

To increase screening efficiency and to address the problem of *C. albicans* persisters, Qiang and colleagues developed a new, high-throughput drug screening system [124]. This system involves microfluidic chips in which *Candida* cells can line up one by one in the microchannels and form biofilms. After treatment, clear images are obtained by fluorescence microscopy and drug efficacy is determined based on the number of surviving persister cells. As 100 compounds can be tested on each chip and 20 chips can be processed at the same time, this system enabled Qiang and coworkers to screen 50,520 small molecules from the Chinese National Compound Library in combination with AmB in one week. In the end, 10 compounds were identified that enhanced AmB’s activity against *C. albicans* biofilm persister cells with more than 30% [124]. To the best of our knowledge, systematic drug repurposing library screens to identify potentiators that enhance antibiofilm activity of antifungal drugs against *Aspergillus* spp. or *Cryptococcus* spp. have not yet been documented.

Furthermore, knowledge regarding previously discovered combination treatments can be exploited in the search for potentiators. Chen and colleagues developed several tools to predict possible synergistic combination treatments, like an Antifungal Synergistic Drug Combination Database (ASDCD), which contains already published synergistic antifungal combinations, targets and other relevant information [125]. Furthermore, they developed an algorithm named “Network-based Laplacian regularized Least Square Synergistic drug combination predication” (NLLS), using data such as previously discovered synergistic combinations and drug-target interactions for its prediction of possible synergistic interactions [119]. Combining these tools led to the identification of lovastatin as a potentiator, acting synergistically with itraconazole against *C. albicans* cells in planktonic and biofilm state [126].

Finally, the identification of potentiators can also be linked to the inhibition of known virulence factors or tolerance mechanisms of fungal biofilms. In this way, Yu and colleagues found that verapamil, a known calcium channel blocker, acts synergistically with fluconazole against *C. albicans* biofilms [127]. Since calcium channels and pumps are key components for *C. albicans*’ virulence, stress response and morphogenesis, verapamil was expected to inhibit *C. albicans* biofilms [127]. Furthermore, Lohse and coworkers screened protease inhibitor libraries to identify new potentiators of fluconazole, AmB or caspofungin against *C. albicans* biofilms, knowing that several secreted proteases are of key importance in *C. albicans* biofilm formation and that a synergistic interaction had been observed between aspartyl protease inhibitors and fluconazole/AmB against *C. albicans* planktonic cultures [128,129,130,131]. Indeed, certain aspartyl protease inhibitors were found to enhance the antibiofilm activity of AmB or caspofungin against *C. albicans* biofilms [131]. In addition, De Cremer and coworkers identified new miconazole potentiators against *C. albicans* biofilms by unraveling miconazole tolerance pathways and determining whether inhibitors of such pathways act synergistically with miconazole against *C. albicans* biofilm cells [120]. They revealed that miconazole treatment resulted in the induction of genes associated with sterol biosynthesis and encoding of drug efflux pumps as well as in downregulation of components of the electron transport chain in *C. albicans* biofilm cells. A synergistic interaction was observed between miconazole and simvastatin, an inhibitor of HMG-CoA reductase which is an important enzyme in the ergosterol biosynthesis. Moreover, a biofilm-specific oxygen-dependent tolerance mechanism was suggested as miconazole acted synergistically with electron transport chain inhibitors against *C. albicans* biofilm cells, but not in oxygen-deprived conditions or against planktonic cultures [120]. Gaining insights into combination treatments’ mode of action is relevant as it may facilitate the discovery of new antibiofilm therapies with a similar mode of action or enable optimization of existing ones [132,133,134]. Therefore, the next section provides an overview of promising in vitro antibiofilm combination treatments and their mode of action. Table 1 summarizes all antibiofilm combination treatments that are mentioned in this review.

## 4. The Mode of Action of Antibiofilm Combinations

Various in vitro studies have been performed to unravel the mode of action of antibiofilm combination treatments. Apparently, antibiofilm combinations target a large variety of processes or components of fungal biofilms, and a single combination is frequently characterized by multiple mechanisms of action. Below, the two main antibiofilm modes of action of combination treatment being targeting of virulence factors and/or inhibition of biofilm-specific drug tolerance mechanisms, are further elaborated.

### 4.1. Antibiofilm Combinations Targeting Virulence Factors

It is difficult to provide a clear definition of fungal virulence factors as some fungal traits are important during infection of the human host as well as in the environment or in benign conditions. Therefore, various different definitions can be found in literature. In general, traits essential for the pathogen to survive or grow in the human host are termed virulence factors [159]. According to the damage-response framework of microbial pathogenesis, a theory conceptualizing host-microbe interactions, virulence factors are microbial components that cause damage to the host [160,161,162,163] and this definition has been mentioned in various studies ever since [164,165,166]. Moreover, Hogan and coworkers described virulence factors as “any factor that a fungus possesses that increases its virulence in the host” [167], a definition that is still cited in more recent work [168]. Here, we provide an overview of combination treatments that target components of the most common fungal genera involved in human disease that contribute to disease development in the host as well as to the organism’s virulence.

#### 4.1.1. Combinations Targeting Biofilm-Specific Structures

Biofilm formation is considered the most important virulence factor of fungal species from the genera *Candida*, *Aspergillus* and *Cryptococcus*. During *C. albicans* biofilm formation for example, yeast cells adhere to a surface, followed by cell proliferation, filamentation/morphological changes and the formation of an extracellular polymer matrix [169]. Morphological transitions from budding yeast cells to filamentous forms, more specific pseudohyphae and hyphae, play an important role in biofilm formation as hyphae are mature biofilms’ main structural components. Their strong capacity to attach to other cells provides integrity and stability to the biofilm structure [170]. Various important adhesins are present on hyphal cell walls, such as Agglutinin-like sequence 3 (Als3) and Hyphal wall protein 1 (Hwp1), which are of key importance for *C. albicans* biofilm adhesion/formation [171,172]. Moreover, various genes are essential for both *C. albicans’* biofilm formation and hyphal development in planktonic cultures, indicating that these processes are controlled by some common regulators, of which transcriptional regulator Enhanced filamentous growth protein 1 (Efg1) is an example [173,174]. Differences in biofilm formation have been observed between various *Candida* spp. For example, *C. albicans* and *Candida tropicalis* form hyphae during biofilm development in contrast to *C. glabrata,* which only forms pseudohyphae in some specific conditions, such as carbon dioxide exposure [169,175,176]. Hence, hyphal formation is a prominent virulence factor, facilitating host tissue invasion during infection and providing strong adhesion to epithelial cells and biofilm stability, in some *Candida* spp. Biofilms of *Candida* spp. that do not form hyphae, consist of yeast layers surrounded by extracellular polymeric substance and are often less virulent [177].

A combination’s effect on these morphological transformations is usually investigated by microscopic observations of hyphal quantity, length or percentage of hyphal formation after incubation of *Candida* cultures in a hyphae-inducing medium, often RPMI 1640 [178], and removal of non-adherent planktonic cells, a method described by Haque and coworkers [179]. Research from Li and colleagues indicated that d-penicillamine, commonly used as a treatment for Wilson’s disease, acts synergistically with fluconazole against *C. albicans* biofilms and planktonic cultures [139]. This combination inhibits the morphological transformation as shorter and fewer hyphae were observed upon combination treatment compared to single compound treatments. Additionally, combination treatment resulted in a reduced intracellular calcium concentration and metacaspase activation, which is linked to apoptosis [139]. Furthermore, a synergistic interaction between AmB and lactoferrin, an iron-chelating milk protein, against various yeasts has been demonstrated [154]. Fernandes and colleagues revealed that lactoferrin combined with AmB synergistically inhibited biofilm formation of *C. glabrata* and *C. albicans.* Moreover, treatment of *C. albicans* with this combination resulted in inhibited hyphal development [154].

Another important aspect of biofilms, providing increased tolerance to antifungals and protection from the immune system, is the extracellular polymer matrix produced by biofilm cells [180,181,182]. Matrix composition differs between in vitro and in vivo situations as the matrix contains up to 98% of host components in vivo [183,184]. Moreover, composition of the matrix and chemistry within the biofilm matrix varies depending on the producing organism [65,185]. The matrix of *C. albicans* biofilms, in which hyphae are embedded, is most intensively studied and consists mainly of proteins (55%) and carbohydrates (25%) with smaller fractions of lipids (15%) and nucleic acids (5%) [65]. β-1,3 glucan, an important polysaccharide in *C. albicans*’ cell wall and contributing to biofilm drug tolerance, is only present in the extracellular polymer matrix in small amounts [65,181]. As suggested by various studies, the extracellular polymer matrix provides tolerance of *Candida* spp. to high antifungal concentrations as its polysaccharides sequester various currently used antimycotics, like fluconazole, AmB and anidulafungin [181,186,187]. Moreover, a correlation has been observed between the quantity of matrix polysaccharides and the sensitivity of *Candida* biofilms to the action of disinfectants and oxidative stressors [188]. Furthermore, matrix polysaccharides contribute to immune evasion [9,182]. The extracellular polymer matrix of *A. fumigatus* biofilms contains proteins (40%), carbohydrates (43%), lipids (14%), aromatic-containing compounds (3%) and extracellular DNA (eDNA) [156,185]. Galactosaminogalactan is one of the main matrix polysaccharides in *A. fumigatus* and is, such as other carbohydrates from the extracellular polymer matrix and cell wall, important for *A. fumigatus*’ ability to adhere to surfaces, although the exact mechanism remains to be elucidated [189,190]. A recent study, however, indicated the importance of deacetylation of galactosaminogalactan for its adherence abilities [191]. Moreover this matrix polysaccharide contributes to immune evasion as it provides protection against neutrophil attacks [192]. Some antibiofilm combination treatments have been discovered that target (compounds of) this protective and biofilm-specific extracellular polymer matrix.

Martins and colleagues focused on eDNA as an interesting target for combination treatment against *C. albicans* biofilms [155]. They previously demonstrated that, although it represents only a small percentage of the extracellular polymer matrix, eDNA plays an important role in the structural integrity of a biofilm [193]. Therefore, deoxyribonuclease I (DNase) was combined with AmB to treat *C. albicans* biofilms in vitro [155]. An improved AmB efficacy in the presence of DNase was observed. However, treatment of *C. albicans* biofilms with a combination of DNase and caspofungin only resulted in a decreased mitochondrial activity as measured by XTT staining and DNase did not increase fluconazole susceptibility of *C. albicans* biofilm cells [155]. Rajendran and coworkers demonstrated the presence of eDNA in the extracellular polymer matrix of *A. fumigatus* biofilms and its importance regarding structural integrity [156]. More specifically, they found that DNase treatment resulted in architectural instability of the biofilms as observed by for example confocal laser scanning microscopy. eDNA release was identified as an antifungal tolerance mechanism in mature *A. fumigatus* biofilms. Improved AmB and caspofungin antibiofilm activity could be achieved in the presence of DNase against *A. fumigatus* [156].

Furthermore, Papi and colleagues used atomic force spectroscopy to demonstrate that treatment of *A. fumigatus* biofilms with the enzyme alginate lyase, which is capable of reducing negatively charged alginate levels in microbial biofilms, resulted in a decrease of adhesion forces within the biofilm [194]. This decrease is indicative for loss of the extracellular polymer matrix [194]. As alginate lyase was capable of degrading matrix polysaccharides of *A. fumigatus* biofilms, Bugli and coworkers combined this enzyme with AmB or its liposomal formulation and observed synergy against *A. fumigatus* biofilms in most cases [94]. This synergy was biofilm-specific. Physical changes observed using atomic force microscopy suggested that alginate lyase may increase the activity of AmB against *A. fumigatus* biofilms by disruption of the extracellular polymeric substances in which the hyphae are embedded, thereby reducing AmB sequestration in the polymer matrix and enabling the drug to reach the biofilm cells [94]. Hence, enzymes degrading components of fungal biofilms’ extracellular polymer matrix are promising potentiators for existing antifungal drugs.

#### 4.1.2. Targeting the Activity or Secretion of Degradative Enzymes

*C. albicans*, *A. fumigatus* and *C. neoformans* all secrete hydrolytic enzymes, that damage the host tissue and facilitate its colonization and invasion [195]. The most important secreted hydrolytic enzymes that contribute to virulence of *C. albicans* are phospholipases, proteinases and lipases [196]. A constitutive expression of genes from the *PLB*, *SAP* and *LIP* gene families, encoding phospholipase B, secreted aspartyl proteinases (Saps) and lipases, respectively, has been observed in *C. albicans* biofilms on mucosal and abiotic surfaces [197,198]. Moreover, Nailis and coworkers demonstrated that a majority of *LIP* and *SAP* genes are upregulated in *C. albicans* biofilms [197]. Furthermore, a correlation has been discovered between the production of Saps and biofilm formation in *C. albicans* as well as between Sap production and adhesion [199,200]. Kadry and colleagues confirmed that biofilm formation is mediated by *SAP* genes (*SAP9* and *SAP10*), as they observed a correlation between the prevalence of Sap9, Sap10 and biofilm formation [201,202,203]. The effects of antibiofilm combinations on the activity or secretion of degradative enzymes is a common subject of investigation in mode of action studies. Extracellular phospholipases contribute to *C. albicans’* virulence by damaging the host’s cell membranes [204]. The effects of a combination on phospholipase activity are usually investigated by the egg yolk agar plate method as described in [205], often with modifications [140,141,143,200]. In general, treated yeast suspension is added to egg yolk agar medium, followed by an incubation period. Phospholipase activity is determined by the equation P_z_ = colony diameter/(colony diameter + precipitation zone diameter) and considered either negative (Pz = 1); very low (Pz = 0.90 to 0.99); low (Pz = 0.80 to 0.89); high (Pz = 0.70 to 0.79), or very high (Pz ≤ 0.69) [140,141,142]. Saps on the other hand contribute to *C. albicans’* virulence by degrading host tissue components and inhibiting phagocytosis-inducing inflammatory reactions, thereby enhancing colonization and tissue invasion [206]. Usually, a reverse transcriptional quantitative PCR (RT-qPCR) is performed to determine whether secreted aspartyl proteinase-related genes (*SAP* genes) are differentially expressed upon combination treatment as compared to single compound or control treatments [140,142].

Gu and coworkers revealed that fluoxetine, an antidepressant from the selective serotonin re-uptake inhibitor class, has a synergistic effect in combination with various azoles against planktonic cultures and early stage biofilms of azole-resistant *C. albicans* isolates, but not against those of non-*albicans Candida* spp. [140]. Treatment of *C. albicans* with this combination resulted in reduced secreted phospholipase activity as well as downregulation of various *SAP* genes [140]. As shown by Zhang and colleagues the antiviral drug ribavirin acts synergistically with fluconazole against azole-susceptible *C. albicans* isolates in all stages of biofilm formation and against azole-resistant *C. albicans* biofilms in early stage of biofilm development [141]. Apparently, ribavirin-fluconazole treatment results in a reduced activity of extracellular phospholipases. Moreover, ribavirin-fluconazole inhibits biofilm formation and hyphal development in azole-susceptible and azole-resistant *C. albicans* cultures. Shorter hyphae and almost no hyphal aggregation were observed upon combination treatment compared to single compound treatments and the negative control [141]. The synergistic combination of fluconazole and licofelone, a dual mPGES-1/LOX inhibitor enzyme involved in the biosynthesis of prostaglandin E2, affects secretion of degradative enzymes as combination treatment resulted in a lowered expression of extracellular phospholipase genes and a reduced activity of Saps [142]. Furthermore, it inhibits biofilm formation of azole-resistant and azole-sensitive *C. albicans* as well as its morphological transition from yeast to hyphal form. However, a synergistic effect of the combination was observed against biofilms in early developmental stages, but not against mature biofilms. In addition, genes that are biofilm-specific or involved in the RAS/cAMP/PKA pathway were expressed at lower levels upon combination treatment [142].

#### 4.1.3. Antibiofilm Combinations Targeting Adhesins

*Candida* spp., *Aspergillus* spp. and *Cryptococcus* spp. are characterized by the presence of adhesins, proteins in the cell wall that mediate adherence of the cells to biotic or abiotic surfaces. These proteins differ between different genera and even between different *Candida* spp. [207]. The Als family, consisting of 8 large cell surface glycoproteins (Als1-Als7, Als9), and the Hwp family of adhesins (Hwp1, Hwp2, Rbt1) are important groups of adhesins in *C. albicans* [207,208]. As demonstrated by Nailis and colleagues *HWP1* as well as the majority of *ALS* genes are upregulated in *C. albicans* biofilms [197]. It is possible that an antibiofilm combination affects these adhesins, thereby interfering with the organism’s ability to attach to surfaces, which is the starting point of biofilm formation. To investigate whether this is the case, an RT-qPCR can be performed, which reveals whether *ALS* genes or *HWP* genes are differentially expressed upon treatment of *C. albicans* cultures with the combination as compared to single compound treatments. That way, Yu et al. found that treatment of *C. albicans* cultures with calcium channel blocker verapamil and fluconazole, a combination with synergistic effects on *C. albicans* biofilm formation and pre-formed biofilms, resulted in a significantly decreased transcription of *ALS3* [127]. This gene encodes the adhesin Als3, a protein that plays a key role in adherence to the host, biofilm development and iron acquisition in *C. albicans* [171]. Moreover, inhibition of biofilm formation and filamentation was also observed upon treatment of *C. albicans* with verapamil-fluconazole as compared to single compound treatments [127]. The combination of ketoconazole with probiotic *Bifidobacterium bifidum* is another example of a synergistic combination that reduces the expression of *ALS* genes in *C. albicans* isolates from oral samples, thereby reducing biofilm development [135].

#### 4.1.4. Modulation of Quorum Sensing by Antibiofilm Combinations

Another interesting target of antibiofilm combination therapy is quorum sensing, which is a communication mechanism between microbial cells that regulates group behaviors like biofilm formation, virulence factor secretion and growth of hyphae [209]. In *Candida* spp., especially in *C. albicans,* the sesquiterpene alcohol farnesol is an important quorum sensing molecule that is able to inhibit biofilm formation as well as *Candida*’s switch from budding yeast to its pseudohyphal or hyphal form at high cell densities [210,211,212]. Since various studies indicated that quorum sensing dependent biofilm formation is inhibited by dietary phytochemicals in several pathogenic bacteria [213,214,215,216,217], Singh and coworkers investigated whether an ethanolic extract of the lichen *U. longissima* could potentiate fluconazole against resistant *C. albicans* [143]. Indeed, potentiation was observed and the dietary flavonoid quercetin appeared to be the potentiating agent. Apparently, quercetin treatment resulted in increased farnesol production, which led to the suppression of biofilm formation and hyphal development and inhibition of proteinase, phospholipase, esterase and hemolytic activity in *C. albicans* cultures. Furthermore, increased farnesol levels due to the presence of quercetin resulted in the induction of fluconazole-mediated apoptosis [143].

### 4.2. Antibiofilm Combinations Targeting Tolerance Mechanisms

In literature, resistance and tolerance are often considered synonyms. However, a separate definition is given in various articles. In general, an organism is termed resistant to an antifungal if it has a reduced susceptibility, more specific an increased minimal inhibitory concentration (MIC), to the antifungal [218,219]. Tolerant organisms are susceptible to the drug, but withstand killing at supra MIC antifungal concentrations under certain conditions (e.g., as in a biofilm) and are even able to grow slowly at those concentrations [218,219]. As shown by Ramage et al., increased activity of drug efflux pumps is a major azole tolerance mechanism in *C. albicans* planktonic cultures as deletion of the most prevalent efflux pumps conferred hypersensitivity to fluconazole [220]. However, increased efflux pump activity likely results in increased antifungal tolerance in early biofilm developmental phases as well, but contributes minimally to antifungal tolerance in mature biofilms [221]. In contrast to tolerance of planktonic *C. albicans* cultures, biofilm tolerance is complex, involving various mechanisms, many of which are biofilm-specific. Targeting these mechanisms is therefore an interesting approach to tackle the problem of biofilm-based fungal infections.

#### 4.2.1. Antibiofilm Combinations Targeting Drug Efflux Pumps

Combination treatments may counteract the increased efflux pump activity, causing drug tolerance in planktonic *Candida* cultures and biofilms in an early developmental stage. Usually, an RT-qPCR is performed to investigate the expression levels of genes encoding efflux pumps in fungal cultures upon combination treatment as compared to single compound treatment or the untreated control. In *C. albicans,* expression of *CDR1*, *CDR2* and *MDR1*, *FLU1* is often determined as these genes encode efflux pumps of the ABC superfamilies and MFS pumps, respectively, which are the main azole efflux pumps of this yeast [137,145,147,222]. Additionally, the red fluorescent dye Rhodamine 6G [223] is frequently used to measure yeast efflux pump activity as this dye uses the same transporters as azoles in yeast [137,144,145,147]. In this assay, cells are incubated in the presence of rhodamine 6G, which is taken up by the cells in starvation conditions, and the tested compounds. After a period of efflux, the remaining fluorescence of intracellular rhodamine 6G as compared to the culture before treatment can be measured by flow cytometry or rhodamine 6G fluorescence in the supernatants can be determined using a microplate reader or a microscope [137,144,145,147]. Alternatively, similar assays using other dyes, like Rh123 or Nile Red efflux assays, can be performed [224,225].

As described by Wang and coworkers the berberine alkaloid palmatine (a plant metabolite) acts synergistically with the triazoles fluconazole and itraconazole against planktonic cultures and biofilms of various *Candida* spp., including *C. albicans*, *C. glabrata* and *Candida krusei* [137]. Upon palmatine-fluconazole treatment, an increased rhodamine G6 accumulation within the cells was observed as well as inhibition of efflux pump associated gene expression. However, variable inhibition levels were observed among different *Candida* spp. as they are characterized by different efflux pump types contributing differently to azole tolerance [137]. A biofilm-active combination can affect both virulence factors and tolerance mechanisms simultaneously. Through screening of a drug repositioning library, Eldesouky and colleagues identified antihyperlipidemic statin drugs as novel fluconazole potentiators against azole-resistant *C. albicans* and pitavastatin was the most promising hit [144]. Although discovered in a planktonic setup, the pitavastatin-fluconazole combination appeared to reduce biofilm formation of *C. albicans* isolates, *C. glabrata* isolates and isolates of emerging pathogen *C. auris.* However, this combination was not active against mature *C. albicans* biofilms. Apparently, pitavastatin interfered with ABC transporter efflux pumps [144]. Furthermore, a synergistic effect of dexamethasone and fluconazole against fluconazole-resistant *C. albicans* planktonic cultures and biofilms in early state of biofilm development has been observed [145]. Dexamethasone is a corticoid, used to treat diverse conditions, like rheumatic problems [226], allergic reactions [227] and as recently identified, COVID-19 [228]. Its combination with fluconazole probably affects *C. albicans’* virulence by reducing extracellular phospholipase activity as well as *C. albicans’* azole tolerance, since inhibition of drug efflux and reduced expression levels of efflux pump associated genes were observed [145]. The combination of the antibiotic gentamicin and fluconazole probably has a similar mode of synergistic action against biofilms of azole susceptible—and resistant *C. albicans* strains in early stages of biofilm development. Here, synergy was also observed against planktonic cultures of resistant *C. albicans,* but not against mature *C. albicans* biofilms or against planktonic cultures of susceptible *C. albicans* isolates or non-*albicans Candida* spp. [146].

The synergistic combination of budesonide, a corticosteroid used to treat symptoms of for example asthma, and fluconazole against biofilms of fluconazole-resistant *C. albicans* is characterized by various modes of action [147]. This combination acts synergistically against mature biofilms, but synergy was weaker as compared to biofilms in early developmental stages. Budesonide-fluconazole interferes with virulence factors, more specific biofilm formation and extracellular phospholipase activity, as well as tolerance mechanisms as budesonide inhibits drug efflux. Moreover, this combination induces apoptosis [147]. The synergistic combination of fluconazole and proton pump inhibitors, often used to treat acid-related diseases, has a similar mode of action against planktonic cultures of fluconazole-resistant *C. albicans* and biofilms in early developmental stages [148]. More specific, proton pump inhibitors suppress drug efflux and in combination with fluconazole, they synergistically inhibit phospholipase activity as well as hyphal development. However, no synergy was observed against mature biofilms [148]. As mentioned by Lu and coworkers [148] different studies concerning the combination of proton pump inhibitors with fluconazole against *Candida* spp. resulted in different outcomes. More specifically, no in vitro synergy was observed against azole-sensitive *C. albicans* in some studies [229,230,231], but fluconazole potentiation by proton pump inhibitor BM2 against fluconazole-resistant *C. albicans* and *Candida dubliniensis* has been demonstrated in others [232,233]. A final example of a combination treatment acting synergistically against planktonic cultures and early-stage biofilms of fluconazole-resistant *C. albicans* by suppressing drug efflux is the combination of fluconazole with gypenosides, triterpenoid saponins from the herbaceous climbing vine *Gynostemma pentaphyllum Makino* [149]. However, gypenosides and fluconazole showed indifferent interactions against fluconazole-sensitive *C. albicans* strains. In addition to drug efflux inhibition, the combination inhibits early stages of biofilm formation as well as *C. albicans*’ morphological transition from budding yeast to its hyphal form. However, further investigation will be necessary to confirm these observations [149]. In general, increased efflux pump activity contributes only minimally to antifungal drug tolerance in mature *C. albicans* biofilms, explaining the poor activity of combinations that inhibit drug efflux against mature *Candida* biofilms.

#### 4.2.2. Cell Membranes or Sterol Biosynthesis Pathways as a Target for Antibiofilm Combinations

The amount of ergosterol in the cell membranes of *C. albicans* biofilm cells is lower than in that of planktonic cells, with much lower levels in mature biofilm cells as compared to early developed biofilm cells [221]. The cell membranes in intermediate/mature biofilms have the most pronounced difference in sterol composition, containing larger amounts of non-ergosterol sterols, like zymosterol [221]. These findings indicate that maintenance of membrane fluidity in mature biofilms depends less on ergosterol, which provides a possible explanation for the limited efficacy of ergosterol-targeting antimycotics, like azoles and polyenes [234]. Thus, the alteration of membrane sterols is an important tolerance mechanism in *C. albicans* biofilm developmental stages in which increased efflux pump activity is not significantly contributing to drug tolerance. Moreover, elevated transcription of genes involved in ergosterol biosynthesis, *ERG11* and *ERG25*, has been observed in *C. albicans* biofilms compared to their planktonic counterparts [235]. Furthermore, inactivation of sterol Δ5,6-desaturase, an enzyme encoded by *ERG3* and playing a key role in ergosterol biosynthesis, results in azole-resistance in *C. albicans* as this enzyme converts nontoxic sterol intermediates, accumulated upon azole treatment, into a toxic sterol [236,237]. To study the effects of a combination treatment on sterol biosynthesis or sterol composition of the plasma membrane, a transcriptional analysis, assessing whether genes involved in ergosterol biosynthesis are differentially regulated, is usually performed [117,126]. In addition, a lipid or sterol extraction [107,117] can be carried out to compare the amounts of certain lipids or sterols within the cells upon combination treatment to that of the control treatments.

Zhou and colleagues observed a synergistic interaction between the statin lovastatin, used to treat high blood cholesterol, and itraconazole against biofilms and planktonic cultures of *C. albicans* [126]. Apparently, this combination acts synergistically by regulating the expression of various genes involved in the ergosterol biosynthesis pathway [126]. Furthermore, the triazoles fluconazole and itraconazole can be potentiated by the anesthetic drug ketamine against planktonic cultures and biofilms of various *Candida* spp. [138]. An altered membrane integrity was observed upon combination treatment and, in addition, this combination induced ROS generation, DNA damage and phosphatidylserine externalization [138]. Lafleur and colleagues identified 2-adamantanamine (AC17), a structural analog of the antiviral drug amantadine, as a novel azole potentiator against *C. albicans* biofilms [117]. Transcriptional profiling indicated that the ergosterol pathway was affected by AC17 as well as filamentation. Furthermore, a decreased amount of ergosterol in cells treated with AC17 as compared to the DMSO control treatment was observed [117]. Various studies have indicated that defective filamentation/hyphal formation and decreased ergosterol levels can co-occur. For example, various ergosterol mutants have been discovered that are defective in filamentation [238,239] and hyphal cells have higher ergosterol levels [240], pointing to possible serious effects on filamentation upon partial inhibition of the ergosterol biosynthesis pathway. Through lipid extraction, Lafleur and coworkers demonstrated that the accumulation of sterol intermediate lanosterol upon voriconazole treatment was abrogated upon combination treatment with voriconazole and AC17 [117]. They therefore suggested that AC17’s target in the ergosterol biosynthesis pathway lies upstream from that of azoles, and that inhibition of both steps of this pathway simultaneously results in synergy. As stated by Lafleur and coworkers, various synergistic combinations that inhibit different parts of the ergosterol pathway exist, such as terbinafine combined with azoles, targeting squalene epoxidation (Erg1p) and α-14 lanosterol demethylase (Erg11p), respectively [117,241,242]. Thus, the combination of AC17 and azoles affects both the ergosterol pathway and hyphal formation, an important virulence factor [117]. Furthermore, diorcinol D, a product isolated from an endolichenic fungus, is capable of potentiating fluconazole against azole-resistant and azole-sensitive *C. albicans* planktonic cultures and mature biofilms and has a fungicidal activity against *C. albicans* on its own [107]. Li and coworkers found that diorcinol D treatment results in a decrease in intracellular sterol content, thereby enhancing azole antifungal action [107]. Moreover, diorcinol D interferes with the expression of *CYP51*, an important gene involved in the ergosterol biosynthesis. In addition, combination treatment resulted in the inhibition of drug efflux pump activity. Hence, diorcinol D-fluconazole targets the ergosterol biosynthesis and drug efflux pump activity simultaneously [107].

#### 4.2.3. Combinations Targeting Stress Response Pathways

Various stress responses are activated during biofilm formation in *Candida* spp., conferring tolerance to antifungals. The cell integrity pathway, which is activated by cell wall stress and in which the enzyme MAPK Mkc1 plays a key role, is an example of such a stress response pathway [243]. This pathway is important for virulence of *Candida* spp. as well as for biofilm formation, hyphal development and biofilm-specific drug tolerance as *mkc1*-null mutants formed abnormal, more susceptible biofilms with reduced hyphal formation as compared to wildtype biofilms [243]. Furthermore, the Ca^2+^ calmodulin-activated serine/threonine-specific protein phosphatase calcineurin is an important component in stress responses conferring biofilm tolerance, since interference with the calcineurin pathway by genetic disruption or by using calcineurin inhibitors results in increased fluconazole susceptibility of *C. albicans* biofilms [152]. The calcium signaling pathway in general is important for the mediation of stress responses, drug tolerance, promotion of virulence and cell wall integrity in *Candida* spp. [152,244,245,246]. The calcium signaling pathway consists of various signaling proteins (e.g., calmodulin, calcineurin), channels, transporters and pumps and many of these compounds or interactions are potential targets for antifungal therapy [247]. Moreover, the stress response pathway involving heat shock protein 90 (Hsp90) is important for antifungal tolerance of *Candida* biofilms [136]. In the planktonic state of *C. albicans*, an association between Hsp90 and both calcineurin and Mkc1 has been observed, which leads to stabilization of calcineurin and Mkc1 and maintenance of drug resistance, but this association was not observed during biofilm growth [136,248,249]. These findings point to a distinct stress response pathway involving Hsp90, that contributes to tolerance of *C. albicans* biofilms. Probably, Hsp90 is a regulator of the matrix sequestration pathway, as impairment of the pathway involving Hsp90 results in lower levels of *β*-1,3-glucans in the biofilm matrix and consequently in reduced drug capture and increased drug susceptibility [136]. In *Candida* spp., Hsp90 is considered essential for biofilm dispersal and azole drug tolerance. In *A. fumigatus,* Hsp90 is required for biofilm drug tolerance and its inhibition results in morphological changes, such as augmented production of hyphae and matrix material [136]. Furthermore, both calcineurin and Hsp90 are of key importance for cell wall integrity in *A. fumigatus* [250,251]. Thus, antibiofilm activity of combinations of an Hsp90 inhibitor with an antifungal has been assessed in various studies.

Garzon and colleagues demonstrated that the antipsychotic fluphenazine, a known human calmodulin inhibitor, increases susceptibility of caspofungin-resistant *C. glabrata* to caspofungin [158]. This caspofungin-resistance resulted from a mutation in *FKS2*, a gene encoding dimeric β-1,3-glucan synthase, which is also a frequent cause of echinocandin-resistance in *C. glabrata* in its natural environment. Fungal calmodulin inhibition by the repurposed fluphenazine in combination with caspofungin resulted in reduced thermotolerance, which is known to be controlled by the calmodulin/calcineurin pathway. Furthermore, a reduced ability of the caspofungin-resistant *C. glabrata* to form biofilms upon combination treatment was observed as compared to caspofungin single treatment [158]. In addition, a synergistic inhibition of (azole-resistant) *C. albicans* growth by combining fluconazole with the antibiotic minocycline has been described by Shi and coworkers [150]. Apparently, treatment of *C. albicans* cultures with fluconazole-minocycline results in a time-dependent increase of intracellular calcium levels compared to single compound treatments, thereby disturbing the cellular calcium homeostasis. Moreover, minocycline facilitates fluconazole penetration in the biofilm as shown by an antifungal penetration study. Here, *C. albicans* biofilms were grown on isopore membrane filters, which were placed on a YPD plate with fluconazole. An antibiotic disk on a similar filter was placed on top of the biofilm. After an incubation period, the disk was removed and placed on a fresh YPD plate, containing *C. albicans*. After 24 h at 35 °C, inhibition zones were measured to assess how much fluconazole had penetrated through the *C. albicans* biofilm [150]. Fluconazole and calcineurin inhibitor cyclosporine A is another synergistic combination against *C. albicans* biofilm formation and mature biofilms that acts by disturbing calcium homeostasis [151]. More specific, fluconazole-cyclosporine A treatment resulted in increased amounts of intracellular calcium as compared to single compound treatments as shown by flow cytometry. Moreover, RT-qPCR indicated downregulation of both biofilm-related genes (e.g., *ALS3*, *HWP1*) and genes associated with drug resistance (e.g., *CDR1*) [151]. In addition, a lower cellular surface hydrophobicity, which is positively correlated with adhesion, morphological transition, and important processes of *C. albicans* biofilm formation [252,253,254], has been observed upon combination treatment as compared to single compound treatments or untreated controls [151]. This synergistic combination was already described by Uppuluri et al., who demonstrated that fluconazole acts synergistically with cyclosporine A as well as with FK506 (tacrolimus), an immunosuppresive drug, against *C. albicans* biofilms [152]. They proved that this synergy resulted from the potentiators’ inhibitory effects on calcineurin [152]. As shown by Khan and colleagues the antifungal and antibiofilm activity of AmB against *C. albicans* can be enhanced by eugenol, the major phenolic compound of clove essential oil [157]. Docking studies showed that eugenol inhibits calcium channels and treatment with this synergistic combination results in decreased levels of intracellular Ca^2+^ in the *C. albicans* cells. Cellular damage, like nuclear DNA damage and a loss of cell wall integrity, was observed using SEM electron microscopy with selective fluorescent probes. This damage was a consequence of the increased ROS production upon combination treatment. Additionally, mitochondrial hyperpolarization was observed and this combination treatment evoked apoptosis [157].

Tu and coworkers revealed that histone deacetylase inhibitor vorinostat acts synergistically with triazoles against planktonic cultures and biofilms of various *Aspergillus* spp. [114]. By performing RT-PCR, they found that treatment with this combination resulted in a decreased expression of both azole-associated multidrug efflux pumps (*MDR1-4*) and *HSP90*. However, they suggest that inhibition of *HSP90* would be the main cause of the observed synergistic effect [114]. Furthermore, Robbins and colleagues found that Hsp90 inhibitor geldanamycin acts synergistically with the echinocandins caspofungin and micafungin and with fluconazole against biofilms of *A. fumigatus* and *C. albicans*, respectively [136]. Geldanamycin also enhanced voriconazole activity against *A. fumigatus* biofilms, but did not improve activity of fluconazole. Microscopy images showed serious damage to the *A. fumigatus* cells and biofilm structure upon voriconazole-geldanamycin or caspofungin-geldanamycin treatment as compared to single compound treatments [136]. Finally, Shekhar-Guturja and colleagues discovered that the depsipeptide beauvericin was capable of enhancing the efficacy of azole antifungals against planktonic cultures of *Saccharomyces cerevisiae* and important human pathogens *C. albicans, A. fumigatus* and *C. neoformans* [153]. Synergy and a high antibiofilm efficacy was observed in *C. albicans*, but was not assessed in other fungal species. As shown in *S. cerevisiae* and *C. albicans*, beauvericin inhibits multidrug efflux as well as TORC1 kinase [153], which is an essential regulator for processes like growth and protein synthesis and which is necessary for the sensing of available nutrients [255]. This way, beauvericin activates protein kinase CK2, resulting in the inhibition of Hsp90, converting the azole’s fungistatic action into fungicidal [153]. Moreover, beauvericin’s interference with the TORC1 complex results in inhibited hyphal formation of *C. albicans* under specific conditions, like increased temperature [256].

These examples point to the potential of the development of antibiofilm combination treatments that target important components of stress response pathways, like calmodulin, calcineurin or Hsp90. However, upon treatment of *C. albicans* with the calmodulin inhibitor fluphenazine, induction of multidrug transporters (Cdr1, Cdr2) has been observed, which could have a negative effect on the efficacy of azoles [257]. Moreover, calcineurin inhibitors, like cyclosporine A, are already used in immunosuppressive therapy in patients receiving a transplant and are insufficiently fungal specific, causing undesired immunosuppressive effects in the human host [258,259]. Similar limitations are observed while using Hsp90 inhibitors in antifungal therapy. For example, Hsp90 inhibitors 17-AAG and 17-DMAG, which are synthetic geldanamycin analogues, are promising anticancer compounds, but low toxic thresholds are observed due to a lack of specificity towards the fungal target [259,260]. Thus, combinations targeting components of stress response pathways show great potential to be developed into novel antibiofilm therapies, if they can be modified to increase specificity for the fungal target [259].

Since fluconazole is the most commonly used antifungal drug, a schematic representation of the mode of action of all fluconazole potentiators against *C. albicans* biofilm cells, discussed in this review, is given in Figure 1.

## 5. Discussion and Conclusions

Increasing evidence points to the involvement of drug tolerant biofilms in mucosal fungal infections, such as vulvovaginal candidiasis [10,22,33,34]. Moreover, fungal biofilms can develop on implantable medical devices, such as catheters, from where fungal cells can enter the bloodstream and spread to other body parts, resulting in life-threatening invasive infections [42,43,44]. Out of the current antifungal armamentarium, only liposomal AmB formulations and echinocandins are effective against those biofilm-based infections [75,76,77].

Combining an antifungal drug with a non-antifungal potentiator to enhance its antibiofilm activity is a novel approach to bypass the limited repertoire of antibiofilm treatments. In this review, we focused on antibiofilm combinations for which the mode of action is known. Obviously, there are many more antibiofilm combination treatments of which the mode of action has not yet (partially) been revealed, or antifungal combinations of which the mode of action is known but that have not (yet) been tested for antibiofilm activity.

The mode of action of a combination treatment against *Candida* spp. is usually investigated in *C. albicans*. Antibiofilm activity against non-*albicans Candida* species is often not assessed, although the amount of non-*albicans Candida* infections is rising [261,262,263]. So far, a wide variety of azole potentiators against *C. albicans* has been identified and smaller amounts of polyene (e.g., AmB) and echinocandin (e.g., caspofungin) potentiators have been described of which the mode of action is known. However, the discovery of novel polyene or echinocandin potentiators would be useful as combination therapy might allow the use of lower, nontoxic antifungal concentrations of these nephrotoxic and embryotoxic agents, respectively [52,67]. Furthermore, various combination treatments against *Aspergillus* spp. have been discovered, but antibiofilm activity was demonstrated for only few of them. Likewise, there is a lack of antibiofilm combination treatments against *Cryptococcus* spp. The biofilm state plays, however, a prominent role during a *C. neoformans* infection as it is important for its survival within macrophages, invasion of the host’s central nervous system as well as host tissue colonization [264]. To the best of our knowledge, antibiofilm combination treatments with a known mode of action against *Cryptococcus* spp. have not yet been described. Other putative virulence factors of *Aspergillus* spp. and *Cryptococcus* spp. exist besides the ones mentioned in this section, which may serve as a target for new antibiofilm combination treatments. For example *A. fumigatus* produces toxins, like gliotoxin, which leads to immunosuppression and damages the host epithelial layer [265,266,267,268,269,270]. *C. neoformans* on the other hand produces melanin, which, for example, provides protection against oxidative damage caused by e.g., oxidants generated by host immune effector cells [271,272].

The next step towards the development of effective antibiofilm combination therapies is the determination of the in vivo efficacy of the most promising in vitro identified antibiofilm combination treatments as well as their performance in clinical trials. Various promising in vitro combination treatments are effective to treat *C. albicans* infections in vivo as assessed in various models (e.g., non-mammalian models, murine models of mucosal, systemic or device-related infections). For example, decreased fungal burden and reduced inflammation of mucosal epithelial cells were observed in a murine vulvovaginal candidiasis model upon fluconazole-quercetin treatment [273]. In the clinic, the capability of combination treatments to treat patients suffering from vulvovaginal candidiasis has been most extensively studied and various potentiators have been discovered that improved azole efficacy to treat this infection [274,275,276]. Furthermore, some in vivo studies have demonstrated increased efficacy of antifungal drugs when combined with a potentiator against *Aspergillus* or *Cryptococcus* infections [154,277,278,279,280]. However, these studies are less abundant compared to those regarding *Candida* infections. To the best of our knowledge, no successful combination treatments, consisting of an antifungal drug and a non-antifungal compound, against *Aspergillus* or *Cryptococcus* infections have been identified in clinical trials so far.

Thorough assessment of in vitro and in vivo compound interactions is crucial as several antagonistic combinations have been identified [281,282]. Moreover, determining interactions between compounds at different ratios/concentrations is very important as compounds may interact differently at varying ratios/concentrations. For example, high concentrations of the antibiotic doxycycline combined with low AmB concentrations showed superior activity against *C. albicans* biofilms compared to single treatment with low AmB concentrations, whereas an antagonistic interaction was observed between a lower doxycycline concentration and low concentrations of AmB [281]. Application of such antagonistic combination treatments could lead to a weaker antibiofilm activity compared to treatment with the most active compound alone and should therefore be avoided. Another risk factor of combination therapy is cytotoxicity, which should be investigated for each individual combination. The goal of using combination treatments is to improve the antifungal drug’s antibiofilm activity against fungal pathogens and to obtain an increased activity at lower concentrations. The use of lower concentrations of, for example, nephrotoxic polyenes (e.g., AmB), is expected to reduce toxicity or side effects [52]. However, it is clear that the assessment of a combination’s effects in clinical trials is necessary for all combination treatments individually to determine whether side effects or interactions unfavorable to the host occur.

Furthermore, efforts have been made to develop novel strategies for an optimal delivery of antifungal combinations in the human body. For example, Sadozai and colleagues created ketoconazole-loaded poly (lactide-co-glycolide) nanoparticles to increase ketoconazole’s bioavailability as these nanoparticles may assemble in wrinkles and hair follicles, resulting in a prolonged ketoconazole release to the skin tissue [283]. This ketoconazole-loaded poly (lactide-co-glycolide) nanoparticles acted synergistically with silver nanoparticles against planktonic *C. albicans* cultures [283]. Therefore, it may be interesting to assess antibiofilm activity of such a combination. Moreover, silver nanoparticles can potentiate fluconazole against early-stage or mature *C. albicans* biofilms [284]. Khan and coworkers improved drug delivery and reduced toxicities by entrapping AmB and fluconazole in fibrin microspheres [285]. They demonstrated an additive potential of the antifungals when entrapped in the microspheres as well as a decreased fungal burden, improved survival and absence of side-effects in *C. neoformans*-infected mice [285]. It might be worth investigating whether such a dual delivery system can be used for an antifungal drug-potentiator antibiofilm combination. Lastly, Thakur et al. proposed fixed dose combination tablets of terbinafine and fluconazole, antifungal drugs that act synergistically against fluconazole-resistant *C. albicans* cultures in vitro, as a better alternative to treat fluconazole-resistant *C. albicans* infections [286]. In the future, fixed dose combination tablets may be developed with an antibiofilm combination consisting of an antifungal drug and a potentiator, if they are compatible with each other as well as the excipients used to make the tablets.

To conclude, combining an antifungal drug with a non-antifungal potentiator may be a valid approach to develop new combination therapies to treat fungal infections, provided that the focus is shifted towards tackling fungal biofilms of various clinically important fungi.

## Figures and Tables

**Figure 1 ijms-21-08873-f001:**
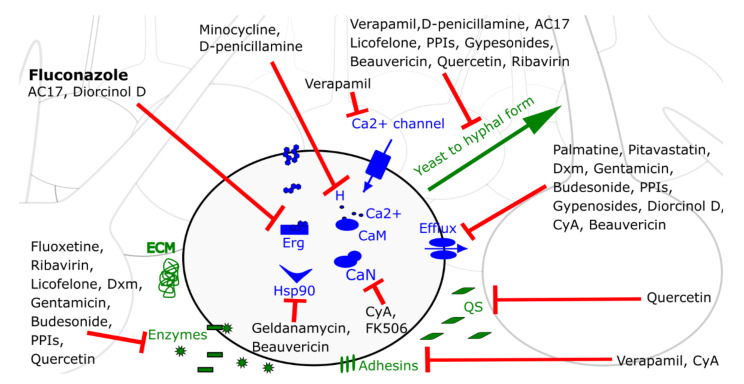
Schematic representation of a *C. albicans* cell in a biofilm context. Important virulence factors and cellular components/pathways involved in *C. albicans* biofilm tolerance are indicated in green and blue, respectively. Fluconazole potentiators are listed with their main modes of action (red arrows), as discussed in this review. *Abbreviations*: Erg, ergosterol biosynthesis pathway; CaM, calmodulin; CaN, calcineurin; QS, quorum sensing; ECM, extracellular polymer matrix; H, calcium homeostasis in general; PPIs, proton pump inhibitors; Dxm, dexamethasone; CyA, cyclosporine A.

**Table ijms-21-08873-t001a:** 

Antifungal Drug	Potentiators	Antibiofilm Activity Spectrum	Targets	Reference
**Azoles**				
Ketoconazole	*Bifidobacterium bifidum*	*C. albicans*	Adhesins (Als)	[135]
Itraconazole	Lovastatin	*C. albicans*	Ergosterol biosynthesis	[126]
Itraconazole Voriconazole Posaconazole	Vorinostat	*A. fumigatus* *A. flavus* *A. terreus*	Drug efflux pumps Hsp90	[114]
Voriconazole	Geldanamycin	*A. fumigatus*	Hsp90	[136]
Miconazole	Artemisinins	*C. albicans*	Not identified	[92]
	Domiphen bromide	*C. albicans* (S & FLC-R) *C. glabrata* *C. auris*	Not identified	[118]
	Simvastatin	*C. albicans*	Ergosterol biosynthesis	[120]
	Antimycin A CCCP Sodium azide	*C. albicans*	Electron transport chain	[120]
Miconazole Voriconazole Fluconazole	2-Adamantanamine	*C. albicans* (S & FLC-R)	Hyphal formation Ergosterol biosynthesis	[117]
Fluconazole Itraconazole	Palmatine	*C. albicans* (S & FLC-R) *C. glabrata* *C. krusei* *C. parapsilosis* *C. tropicalis* *C. guilliermondii*	Drug efflux pumps	[137]
	Ketamine	*C. albicans* (FLC-R)	Membrane integrity ROS production Apoptosis	[138]
Fluconazole	Minocycline	*C. neoformans* (S & R)	Not identified	[115]
	Verapamil	*C. albicans*	Hyphal formation Adhesins (Als3) Calcium channels	[127]
	D-penicillamine	*C. albicans* (S & R)	Hyphal formation Intracellular calcium homeostasis Metacaspase activation	[139]
	Fluoxetine	*C. albicans* (R)	Secreted phospholipases & aspartyl proteinases	[140]
	Ribavirin	*C. albicans* (S & R)	Hyphal formation Secreted phospholipases	[141]
	Licofelone	*C. albicans* (S & R)	Hyphal formation Secreted phospholipases & aspartyl proteinases RAS/cAMP/PKA Pathway	[142]
	Quercetin	*C. albicans* (R)	Quorum sensing Hyphal formation Secreted phospholipases & proteinases	[143]
	Pitavastatin	*C. albicans* (R) *C. glabrata* *C. auris*	Drug efflux pumps	[144]
	Dexamethasone	*C. albicans* (R)	Secreted phospholipases Drug efflux pumps	[145]
	Gentamicin	*C. albicans* (S & R)	Secreted phospholipases Drug efflux pumps	[146]
	Budesonide	*C. albicans* (R)	Secreted phospholipases Drug efflux pumps Apoptosis induction	[147]
	Proton pump inhibitors (e.g., omeprazole, rabeprazole)	*C. albicans* (R)	Hyphal formation Secreted phospholipases Drug efflux pumps	[148]
	Gypenosides	*C. albicans* (R)	Hyphal formation Drug efflux pumps	[149]
	Diorcinol D	*C. albicans* (S & R)	Drug efflux pumps Ergosterol biosynthesis	[107]
	Minocycline	*C. albicans* (S & R)	Calcium homeostasis Penetration into biofilm	[150]
	Cyclosporine A	*C. albicans*	Adhesins (Als3, Hwp1) Drug efflux pumps Calcineurin Cellular surface hydrophobicity	[151,152]
	FK506 (Tacrolimus)	*C. albicans*	Calcineurin	[152]
	Geldanamycin	*C. albicans*	Hsp90	[136]
	Beauvericin	*C. albicans*	Hyphal formation Drug efflux pumps Hsp90 (via TORC1 kinase & CK2 kinase)	[153]
**Polyenes**				
AmB	Aspirin	*C. albicans* *C. parapsilosis*	Not identified	[105]
	Toremifene citrate Drospirenone, Perhexiline maleate	*C. albicans* *C. glabrata*	Not identified	[116]
	10 small molecule compounds	*C. albicans* persisters	Not identified	[124]
	Aspartyl protease inhibitors	*C. albicans*	Aspartyl proteases	[131]
	Lactoferrin	*C. albicans* *C. glabrata*	Hyphal formation	[154]
	Deoxyribonuclease I	*C. albicans* *A. fumigatus*	Extracellular polymer matrix	[155,156]
	Alginate lyase	*A. fumigatus*	Extracellular polymer matrix	[94]
	Eugenol	*C. albicans*	Calcium channels ROS production Apoptosis	[157]
**Echinocandins**				
Caspofungin	Toremifene citrate Drospirenone Perhexiline maleate	*C. albicans* *C. glabrata*	Not identified	[116]
	Aspartyl protease inhibitors	*C. albicans*	Aspartyl proteases	[131]
	Deoxyribonuclease I	*C. albicans* *A. fumigatus*	Extracellular polymer matrix	[155,156]
	Fluphenazine	*C. glabrata* (R)	Calmodulin	[158]
Caspofungin Micafungin	Geldanamycin	*A. fumigatus*	Hsp90	[136]

^1^*C. albicans* without an indication regarding susceptibility are common susceptible strains. Abbreviations: S, susceptible strains/isolates; R, strains/isolates resistant to the antifungal drug in question; FLC-R, fluconazole-resistant strains/isolates.

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
