# Peer review of "Combination Therapy to Treat Fungal Biofilm-Based Infections"

_ijms, 2020, doi:10.3390/ijms21228873_

Round 1
Reviewer 1 Report
Dear authors,
“Combination therapy to treat fungal biofilm-based infections” is well-written review and the topic is relevant and interesting. Authors need to pay attention to certain points in the review.
Authors tend to discuss mostly on Candida albicans. It is very important to include studies done in other fungal pathogens as well.
Line 111: Another approach is repurposing of old drugs. What do you mean by old drugs? Please use right terms here.
Line 114: Authors briefly mention about live animal models for Cryptococcus. How about C. elegans interaction with Candida albicans and other fungi? It would be interesting to compare other fungal information here.
Line 119 Drug interactions: synergy
Authors should give some examples of drug synergy in this case. Screening for novel anti-biofilm combinations. It would be interesting to see all possible work has been done in each fungal pathogens in addition to Candida albicans.
Author Response
Point 1: Authors tend to discuss mostly on Candida albicans. It is very important to include studies done in other fungal pathogens as well.
Response 1: The majority of research on the identification and activity of novel antibiofilm combination treatments and their modes of action has been performed on C. albicans. Studies describing antibiofilm combination treatments against other important fungal genera, such as Aspergillus spp. or Cryptococcus spp. are scarce. We have mentioned this now explicitly in the introduction (lines 44-47) and discussion (lines 728-731, 733-734).
To extend the focus of the review more toward other fungal pathogens, we added following info: (i) lines 67-70 discussing in vivo evidence regarding the involvement of biofilms in common Aspergillus infections, such as aspergilloma and invasive pulmonary aspergillosis; (ii) lines 191-193 describing the synergy between the antibiotic minocycline and fluconazole against early stage biofilms of fluconazole-resistant and susceptible C. neoformans strains [1]; (iii) lines 210-214 describing the quaternary ammonium compound domiphen bromide as a miconazole potentiator, not only against biofilms of susceptible C. albicans and of fluconazole-resistant C. albicans isolates, but also against biofilms of intrinsically azole-resistant C. glabrata and of emerging pathogen C. auris [2]; (iv) lines 344-351 providing more detailed information on the composition and function of the extracellular polymer matrix in A. fumigatus biofilms; (v) lines 361-367 describing the discovery of eDNA as a structural component of the extracellular polymer matrix of A. fumigatus, which is part of the antifungal tolerance mechanism in mature biofilms. Improved AmB and caspofungin antibiofilm activity could be achieved in the presence of DNase against A. fumigatus [3]. Combinations described under (ii), (iii) and (v) were additionally added to Table 1.
Point 2: Line 111: Another approach is repurposing of old drugs. What do you mean by old drugs? Please use right terms here.
Response 2: We agree that the term “old drugs” is not appropriate and replaced it by “existing drugs, primarily used to treat conditions other than fungal infections” (lines 127-128).
Point 3: Line 114: Authors briefly mention about live animal models for Cryptococcus. How about C. elegans interaction with Candida albicans and other fungi? It would be interesting to compare other fungal information here.
Response 3: We added lines 130-138 in which we explain that C. elegans is an interesting in vivo model organism to study fungal infections caused by several medically important fungal pathogens. Several virulence factors, such as hyphal formation or the polysaccharide capsule have been identified in C. elegans infected with C. albicans or C. neoformans. Furthermore, various studies describe C. elegans as a model organism that enables high throughput screening for preclinical drug discovery, especially against infection by C. albicans. The use of C. elegans as an in vivo model for A. fumigatus infections is still under investigation, but promising results have recently been reported regarding pathogenicity investigation, screening for new antifungal drugs and assessment of drug efficacy [4].
Point 4: Line 119 Drug interactions: synergy. Authors should give some examples of drug synergy in this case.
Response 4: We have added following examples of drug synergies (lines 185-193): (i) the combination of AmB with anti-inflammatory drug aspirin, which is synergistic against biofilms of C. albicans and C. parapsilosis and is described by Zhou et al. [5] (lines 185-191). (ii) the combination of histone deacetylase inhibitor vorinostat with triazole antifungals as an example of a synergistic combination against A. fumigatus biofilms (lines 185-191). (iii) the synergistic combination of fluconazole and the antibiotic minocycline against early stage biofilms of fluconazole-resistant and susceptible C. neoformans (lines 191-193). Combination treatments described under (i) and (iii) were additionally added to Table 1.
Point 5: Screening for novel anti-biofilm combinations. It would be interesting to see all possible work has been done in each fungal pathogens in addition to Candida albicans.
Response 5: To extend the focus of this section also toward non-albicans Candida spp. and to provide an explanation for the absence of studies that discuss antibiofilm combination treatments against other fungal species such as Aspergillus spp. and Cryptococcus spp., we added following info: (i) lines 210-214 describing the identification of the quaternary ammonium compound domiphen bromide as a miconazole potentiator, not only against biofilms of susceptible C. albicans and of fluconazole-resistant C. albicans isolates, but also against biofilms of intrinsically azole-resistant C. glabrata and of emerging pathogen C. auris,; (ii) lines 232-234 clarifying that, to the best of our knowledge, systematic drug repurposing library screens to identify potentiators that enhance antibiofilm activity of antifungal drugs against Aspergillus spp. or Cryptococcus spp. have not been documented (yet). All available studies regarding antibiofilm combination treatments (consisting of an antifungal drug and a non-antifungal as potentiator) against Aspergillus spp. or Cryptococcus spp. are included in this review.
- Kong, Q.; Cao, Z.; Lv, N.; Zhang, H.; Liu, Y.; Hu, L.; Li, J. Minocycline and Fluconazole Have a Synergistic Effect Against Cryptococcus neoformans Both in vitro and in vivo. Front. Microbiol. 2020, 11, 836, doi:10.3389/fmicb.2020.00836.
- Tits, J.; Cools, F.; De Cremer, K.; De Brucker, K.; Berman, J.; Verbruggen, K.; Gevaert, B.; Cos, P.; Cammue, B.P.A.; Thevissen, K. Combination of Miconazole and Domiphen Bromide Is Fungicidal against Biofilms of Resistant Candida spp. Antimicrob. Agents Chemother. 2020, 64, e01296-20, doi:10.1128/AAC.01296-20.
- Rajendran, R.; Williams, C.; Lappin, D.F.; Millington, O.; Martins, M.; Ramage, G. Extracellular DNA release acts as an antifungal resistance mechanism in mature Aspergillus fumigatus biofilms. Eukaryot. Cell 2013, 12, 420–429, doi:10.1128/EC.00287-12.
- Ahamefule, C.S.; Qin, Q.; Odiba, A.S.; Li, S.; Moneke, A.N.; Ogbonna, J.C.; Jin, C.; Wang, B.; Fang, W. Caenorhabditis elegans-Based Aspergillus fumigatus Infection Model for Evaluating Pathogenicity and Drug Efficacy. Front. Cell. Infect. Microbiol. 2020, 10, 320, doi:10.3389/fcimb.2020.00320.
- Zhou, Y.; Wang, G.; Li, Y.; Liu, Y.; Song, Y.; Zheng, W.; Zhang, N.; Hu, X.; Yan, S.; Jia, J. In vitro interactions between aspirin and amphotericin B against planktonic cells and biofilm cells of Candida albicans and C. parapsilosis. Antimicrob. Agents Chemother. 2012, 56, 3250–3260, doi:10.1128/AAC.06082-11.

Reviewer 2 Report
In the manuscript „Combination therapy to treat fungal biofilm-based infections” authors summarize interestingly the previously published information on the possibility of using combination of drugs in the treatment of infections caused by Candida, Aspergillus and Cryptococcus and associated with the formation of biofilms. The topic is up-to-date and quite important due to the significant problems with treating such infections related to fungal resistance to classic drugs.
The work is well written, it covers broadly given issue.
I would only like to ask the authors to add a paragraph of information about the potential side effects of drug combinations used or possible interactions unfavorable to the host, if such information is available.
Furthermore, is the thickness of red arrows in Figure 1 significant? If not, they should be the same.
I found also a few more minor phrases for improvement:
Line 29 “body sites” instead of “body parts”
Line 377 “large” instead of “big”
Line 378 please change “hwp1, hwp2” to “Hwp1, Hwp2”
Author Response
Point 1: I would only like to ask the authors to add a paragraph of information about the potential side effects of drug combinations used or possible interactions unfavorable to the host, if such information is available.
Response 1: We added a paragraph (lines 756-771) discussing potential side effects of drug combinations. We highlight the importance of in vitro and in vivo investigation of the effects of a combination of two compounds at varying ratios as exemplified by the combination of doxycycline and AmB. Here, high concentrations of the antibiotic doxycycline combined with low AmB concentrations showed superior activity against C. albicans biofilms compared to single treatment with low AmB concentrations, whereas an antagonistic interaction was observed between a lower doxycycline concentration and low concentrations of AmB. Another risk factor of combination therapy is cytotoxicity, which should be investigated for each individual combination. The goal of using combination treatments is to improve the antifungal drug’s antibiofilm activity against fungal pathogens and to obtain an increased activity at lower concentrations. The use of lower concentrations of, for example, nephrotoxic polyenes, is expected to reduce toxicity or side effects. It is clear that the assessment of a combination’s effects in clinical trials is necessary for all combination treatments individually to determine whether side effects or interactions unfavorable to the host occur.
Point 2: Furthermore, is the thickness of red arrows in Figure 1 significant? If not, they should be the same.
Response 2: The thickness of the red arrows in Figure 1 is not significant and we agree that this should be the same for every arrow to avoid confusion. Therefore, all red arrows have the same thickness in revised Figure 1. Additionally, most of the text within the figure has been moved to the sides, resulting in a reduced interference with the drawing and making Figure 1 more clear / comprehensive.
Point 3: Line 29 “body sites” instead of “body parts”
Response 3: “Body parts” has been replaced by “body sites” (line 29).
Point 4: Line 377 “large” instead of “big”
Response 4: “Big” has been replaced by “large” (line 430).
Point 5: Line 378 please change “hwp1, hwp2” to “Hwp1, Hwp2”
Response 5: “hwp1, hwp2” has been replaced by “Hwp1, Hwp2” (line 431).

Reviewer 3 Report
The review presented for review concerns the use of therapeutic methods of yeast infections, in which the formed biofilm plays an important role. The topic undertaken by the authors is very important, and the prepared summary of the current knowledge seems interesting and necessary. The authors focused on the three most important Candida microorganisms, Aspergillus, or Cryptococcus, which are responsible for the majority of infections leading to disease development.
The structure of the article is well-thought-out, divided into chapters presenting the effects of selected therapeutic methods on the molecular mechanism of pathogen virulence. This arrangement guides the reader logically through the problem at hand while extracting the key, interesting information. Besides, important information has been collected in a clear table and summarized with a drawing of the effect of therapy on C. albicans yeast. Due to the multitude of factors, the figure is somewhat unreadable and requires a moment of reflection, which contradicts the idea of a schematic summary of information. However, the information provided is correct and relevant. In the opinion of the reviewer, there was no analogous treatment summary for the other microorganisms discussed, which of course is associated with a small amount of relevant information on a given topic.
The literature review is carried out meticulously, it contains 268 items which contain all relevant information, and, importantly, the latest literature items are discussed. For unknown reasons, there is no mention of a physical therapeutic method based on photodynamic therapy (PDT), where several articles on the use of PDT in combating biofilms have been published. For example: 10.3389/fmicb.2018.01299, 10.1007/978-1-60761-697-9_13, 10.1016/j.pdpdt.2020.101825. I recommend that the authors consider supplementing this information.
One editorial note that I believe deserves revision is the headings of the subsections in Chapter 4, which all start the same: "Antibiofilm combinations targeting ..". Maybe you should consider varying the titles?
The above comments do not affect my assessment of the work, which is high and I agree that it is accepted for publication.
Author Response
Point 1: Due to the multitude of factors, figure 1 is somewhat unreadable and requires a moment of reflection, which contradicts the idea of a schematic summary of information. However, the information provided is correct and relevant.
Response 1: We agree that Figure 1 contains a lot of information, which makes it somewhat difficult to read. Therefore, most of the text has been moved to the sides of the figure. This way, it interferes less with the drawing, making Figure 1 more clear / comprehensive. In contrast to the initially submitted Figure 1, all red arrows have the same thickness in revised Figure 1 as the varying thickness was not significant and could lead to confusion.
Point 2: In the opinion of the reviewer, there was no analogous treatment summary for the other microorganisms discussed, which of course is associated with a small amount of relevant information on a given topic.
Response 2: Indeed, no analogous treatment summery was provided for the other microorganisms discussed due to the small amount of relevant information regarding antibiofilm combination treatments (consisting of an antifungal drug and a non-antifungal potentiator) against those microorganisms. We have mentioned this now more explicitly in the introduction (lines 44-47) and discussion (lines 728-731, 733-734).
Point 3: For unknown reasons, there is no mention of a physical therapeutic method based on photodynamic therapy (PDT), where several articles on the use of PDT in combating biofilms have been published. For example: 10.3389/fmicb.2018.01299, 10.1007/978-1-60761-697-9_13, 10.1016/j.pdpdt.2020.101825. I recommend that the authors consider supplementing this information.
Response 3: We have now included information on antimicrobial photodynamic therapy in our overview of antibiofilm approaches (lines 117-123). We mention that the use of this method can affect virulence factors, such as adhesion or biofilm formation, of fungal pathogens (e.g. C. albicans).
Point 4: One editorial note that I believe deserves revision is the headings of the subsections in Chapter 4, which all start the same: "Antibiofilm combinations targeting ..". Maybe you should consider varying the titles?
Response 4: We agree that varying the titles would make Chapter 4 more attractive. To this end, we left subtitles at heading level 2 as they were, but changed the original subtitles at heading level 3, originally starting by “Antibiofilm combinations targeting…” to the following subtitles:
4.1.1. Combinations targeting biofilm-specific structures
4.1.2. Targeting the activity or secretion of degradative enzymes
4.1.3. Antibiofilm combinations targeting adhesins
4.1.4. Modulation of quorum sensing by antibiofilm combinations
4.2.1. Antibiofilm combinations targeting drug efflux pumps
4.2.2. Cell membranes or sterol biosynthesis pathways as a target for antibiofilm combinations
4.2.3. Combinations targeting stress response pathways
